🔓 | **Open Peer Review** | Human Microbiome | Research Article

# Predicting age from binarized human oral microbial data combined with an ensemble of classifiers

Yuxiang Zhou,[1] Yanyun Wang,[2] Benyang Xiao,[1] Shuangshuang Wang,[1] Zhirui Zhang,[1] Xindi Wang,[1] Bo Liu,[1] Yufei Yang,[1] Chuanxu Wang,[1] Chengye Zhou,[1] Miao Liao,[1] Feng Song,[1] Haibo Luo[1]

**ABSTRACT** It is well established that the composition of the human microbiome changes with age; however, limited research has explored the association between the oral microbiome and aging, as well as its potential for age prediction. In this study, we investigated the correlation between the oral microbiome and age by analyzing samples from 150 individuals across a wide age range (6–78 years). The observed species richness and Chao1 index significantly increased with age. Permutational multivariate analysis of variance (PERMANOVA), using both Bray–Curtis and Jaccard distances, identified age as a major factor influencing microbial variation. After a comprehensive comparison of four different oral microbiome data processing approaches, we then developed an ensemble model based on binarized oral microbial data, incorporating an eXtreme Gradient Boosting (XGBoost) algorithm with 32 classifiers. This ensemble model achieved a mean absolute error (MAE) of 7.20 years in the independent validation set ($n = 15$) and 4.33 years in the 20–59 age subgroup ($n = 12$), significantly outperforming traditional models. When the sample size increased to 2,550, the MAE in the independent validation set ($n = 255$) was reduced to 4.80 years, with the 20–59 age subgroup ($n = 232$) achieving an MAE of 3.76 years, highlighting its generalizability and robustness. Additionally, compared to the previously published model for age prediction based on oral microbiome, our model demonstrated significantly superior performance. These findings support the potential of integrating binarized microbial data with ensemble modeling as a promising direction for human age prediction based on the microbiome.

**IMPORTANCE** PERMANOVA analysis with Jaccard distances revealed age as a major determinant of variation in microbial composition, highlighting the potential of binarized oral microbial data as a novel predictor for human age prediction. Furthermore, we developed an ensemble model combining an XGBoost algorithm and 32 classifiers to predict age from binarized oral microbial data. The model achieved an MAE of 7.20 years in the independent validation set ($n = 15$) and 4.33 years in the 20–59 age subgroup ($n = 12$). When applied to predict age in 2,550 samples from previous studies, the ensemble model outperformed the prior model, achieving an MAE of 4.44 years compared to the previous model's 4.94 years. These findings demonstrate that binarized oral microbial data, along with the ensemble model we developed, can effectively predict human age and provide a solid foundation for future age-related research.

**KEYWORDS** age prediction, binarized oral microbial data, ensemble of classifiers, XGBoost algorithm

**Peer Reviewers** Shi Huang, The University of Hong Kong, Hong Kong, Hong Kong; Aman Ashar, Karachi Medical and Dental College, Karachi, Pakistan; Michael L. Neugent, University of Texas at Dallas, Richardson, Texas, USA

Address correspondence to Feng Song, fengsong9@163.com, or Haibo Luo, luohaibo@263.net.

Yuxiang Zhou and Yanyun Wang contributed equally to this article. The author order was determined by drawing straws.

The authors declare no conflict of interest.

See the funding table on p. 20.

*[This article was published on 31 October 2025 with the incorrect Peer Review History file. The file was corrected in the current version, posted on 10 November 2025.]*

The human microbiome is a highly dynamic and adaptable ecosystem, encompassing bacteria, fungi, archaea, and viruses. Extensive research has demonstrated that the composition, diversity, and function of the microbiome undergo profound changes throughout the lifespan, characterized by rapid shifts during childhood, a prolonged

period of lifestyle-associated stability, and, ultimately, age-related modifications (1–3). The gut microbiome has been the subject of intensive study, particularly in the context of the development of aging clocks based on microbial signatures (4, 5). In contrast, age-related alterations in the oral microbiome remain largely underexplored. Some studies have reported a reduction in the diversity of the salivary microbiome in older adults, with certain bacterial taxa exhibiting increased abundance compared to younger individuals (≤64 years) (6, 7). Moreover, the high prevalence of oral diseases among the elderly suggests that age-related changes in the oral microbiota may play a significant role in the pathogenesis of these conditions (8). These findings highlight the pressing need and feasibility of investigating the composition of the oral microbial community in relation to age in greater depth.

16S rRNA gene-targeted sequencing is a widely recognized and cost-efficient tool for profiling microbial communities (9). However, sequencing data processing typically yields tens of thousands of amplicon sequence variants (ASVs), with relative abundances of each sample expressed as continuous variables ranging from 0 to 1. The resulting datasets are inherently high-dimensional, sparse, and prone to noise, which poses substantial challenges for direct modeling and subsequent analytical interpretation (5). A specialized discretization method—binary discretization—may offer an effective solution to this issue. Binary discretization partitions the range of continuous-valued attributes into only two intervals. For instance, for a continuous attribute A, by identifying a threshold T, the data is divided into two parts: $A \leq T$ (labeled as 0) and $A > T$ (labeled as 1) (10). This approach converts continuous variables into binary variables, simplifying the data and enhancing robustness to outliers. Binary discretization also mitigates potential noise in continuous data, thereby improving its reliability (11). Furthermore, this method is more efficient and straightforward in certain machine learning applications, allowing for clearer identification of relationships between variables when interpreting complex data structures (12). For high-dimensional datasets, such as microbiome data containing numerous ASVs, the application of this technique can improve model construction and performance.

Moreover, selecting the appropriate algorithm is particularly crucial when dealing with large-scale datasets. While simple and intuitive linear models can be effective in some cases, they are limited in their ability to handle collinear variables and model nonlinear relationships (13). These constraints may impede their performance when applied to complex data. To address these challenges, more advanced algorithms, such as Support Vector Machine (SVM), Random Forest (RF), and eXtreme Gradient Boosting (XGBoost), have been developed. These methods are capable of capturing more intricate relationships between variables and are resilient to outliers and noise. However, their effectiveness can vary depending on the dataset, and the complex parameter tuning necessary for optimal performance poses significant challenges for their widespread application (14, 15).

The application of ensemble models, which integrate multiple classifiers, offers a promising solution to improve the accuracy, stability, and generalizability of prediction models. Research has demonstrated that by combining several models, the ensemble model can effectively mitigate the risk of model selection errors and yield a more accurate approximation of the true underlying relationship compared to any single classifier (16). Furthermore, they enhance the model's ability to generalize to new data (17). For instance, the output targets method discretizes the target variable into distinct bins, which are then processed by multiple classifiers to form an ensemble model (17). Specifically, the ensemble model consists of several classifiers, each having different target bins, with adjacent bin boundaries gradually adjusted to cover the entire range of the target variable (18). And the final prediction is determined through a majority vote from the predictions of all classifiers within the ensemble. This approach, which combines discretization with ensemble learning, effectively reduces noise and enhances the model's robustness and applicability, making it particularly suitable for large-scale regression problems (11, 18).

In this study, we have collected saliva samples from 150 individuals across a wide age range (6–78 years). High-throughput sequencing of the 16S rRNA gene V3–V4 regions was performed to profile the oral microbiome composition and explore its relationship with age. Subsequently, we developed an ensemble age prediction model, incorporating 32 XGBoost algorithms based on binarized oral microbial data, and compared its performance with other models and microbial data processing methods. To further assess its robustness, we also applied our model to additional datasets from previous studies. All results collectively indicate that our ensemble model, combined with binarized microbial data, demonstrates strong predictive accuracy and robustness.

## RESULTS

### Oral microbial profiling by age

A total of 150 individuals were included in the study. According to the previous study (19), they were divided into three groups based on their age distribution (young: 0–29 years; middle-aged: 30–64 years; and old: ≥65 years). At the phylum level, the average relative abundances of *Proteobacteria* accounted for 33.04%, as the most predominant bacterial phylum, followed by *Firmicutes* (31.71%), *Bacteroidota* (15.81%), and other 18 phyla (Fig. 1a; Table S1-1). Comparisons using the Kruskal–Wallis test followed by Dunn's post hoc test between the three age groups demonstrated that there existed a significant difference between the groups in the relative abundance of the phyla *Proteobacteria*, *Firmicutes*, *Spirochaetota*, and *Synergistota* (Fig. 1b; Table S1-3), in which the phylum *Proteobacteria* increased with age, whereas *Firmicutes* showed the opposite trend, decreasing with age. Meanwhile, the phylum *Acidobacteriota* only appeared in the young group; the phyla *Bdellovibrionota*, *Gemmatimonadota*, and *Planctomycetota* only occurred in the middle-aged group; and the phylum *Dependentiae* only in the old group. These phyla might serve as promising features for age discrimination (Table S1-2). At the genus level, the most predominant bacterial genus was *Neisseria*, belonging to the phylum *Proteobacteria*, which increased with age, the same as the phylum *Proteobacteria*. Similarly, the genus *Streptococcus* affiliated with the phylum *Firmicutes*, as the second largest bacterial genus, decreased with age, as did the phylum *Firmicutes* (Fig. S1; Table S1-4).

Linear discriminant analysis effect size (LEfSe) was analyzed to reveal 37 taxa that differed among the young, middle-aged, and old groups (LDA >2.0, $P < 0.05$, Fig. 2a and b; Table S2). Figure 2c and d demonstrated the differences in the oral microbes of the young, middle-aged, and old groups at the genus level. The relative abundances of *Neisseria*, *Centipeda*, and *Comamonas* were significantly higher in the old group compared to the young group, as indicated by the Dunn's post hoc test ($P < 0.05$). In addition, the relative abundances of *Neisseria* were also markedly higher in the old group than in the middle-aged group. Meanwhile, the abundances of *Centipeda*, *Comamonas*, *Treponema*, *Fretibacterium*, *Filifactor*, *Mycoplasma*, *[Eubacterium] saphenum group*, *Defluviitaleaceae_UCG-011*, and *Rikenellaceae_RC9_gut_group* were higher in the middle-aged group than in the young group. These results were regarded to explain the difference between the three groups.

Alpha-diversity analysis, based on the observed species richness and Chao1 index, revealed significant differences among the three age groups (Kruskal–Wallis test, $P < 0.05$). Notably, Dunn's post hoc test indicated a particularly pronounced variation between the young and middle-aged groups, with $P$ values for both indices also below 0.05 (Fig. 1c). Principal coordinate analysis (PCoA) based on Bray–Curtis and Jaccard distances revealed differences among the three age groups, which was further confirmed by permutational multivariate analysis of variance (PERMANOVA). Among the five factors (age, sex, smoking, alcohol consumption, and BMI), PERMANOVA indicated that only age and sex had significant effects on the overall structure of the oral microbiota, regardless of whether Bray–Curtis or Jaccard distances were used (Bray–Curtis: age $R^2 = 0.0200$, $P < 0.01$; sex $R^2 = 0.0104$, $P < 0.05$; Jaccard: age $R^2 = 0.0166$, $P < 0.001$; sex $R^2 = 0.0088$, $P < 0.01$) (Fig. 1d and e). Additionally, the Jaccard distance analysis of binarized oral microbial

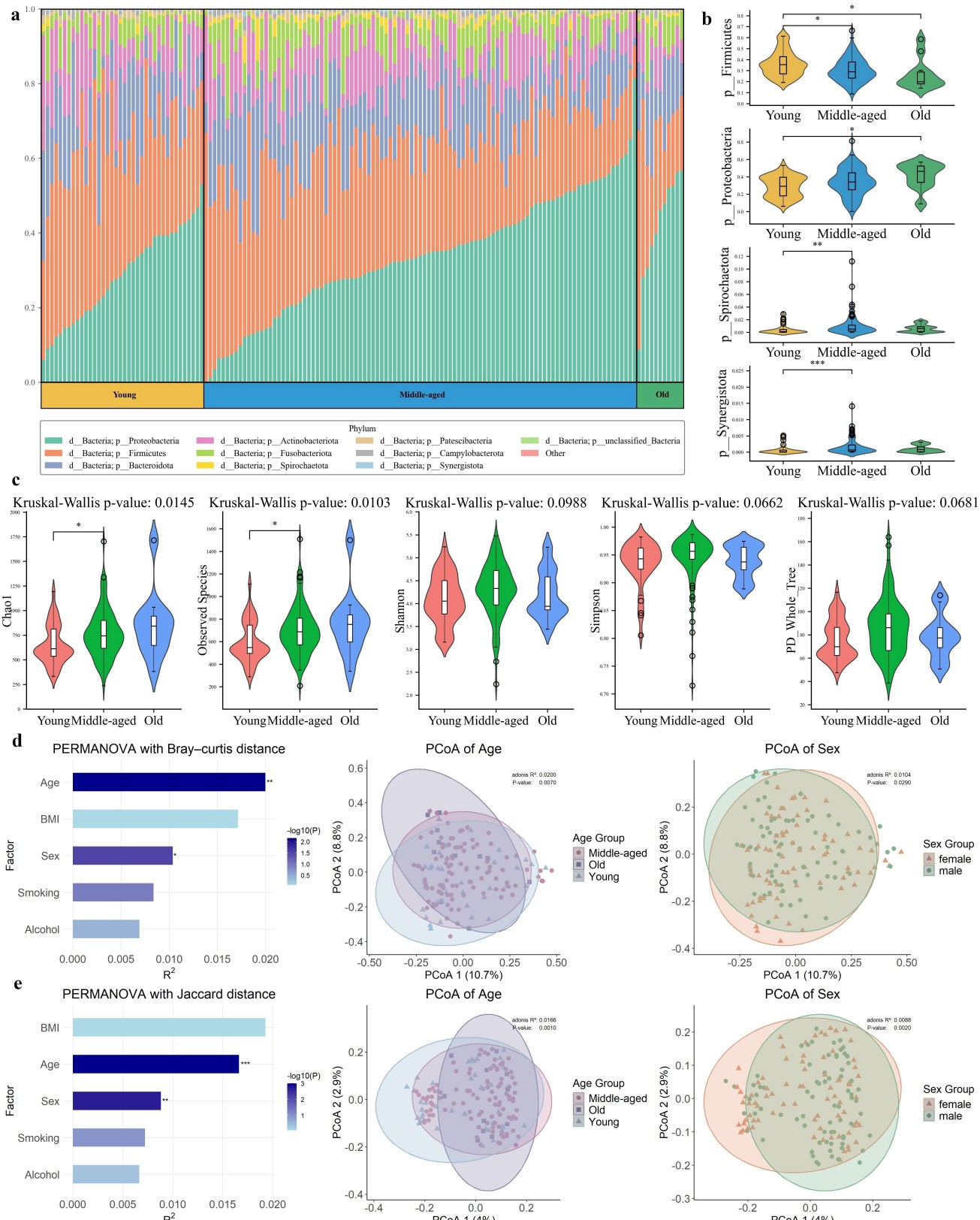

**FIG 1** The analysis of oral microbial profiling based on age. The samples were categorized into three age groups: young (0–29 years), middle-aged (30–64 years), and old (≥ 65 years). (a) Relative abundance of oral bacteria grouped by age at the phylum level. (b) Violin plots of phyla exhibited statistically significant differences (*P* value of Kruskal–Wallis test < 0.05) across the three age groups (*P* value was calculated using the Kruskal–Wallis test followed by Dunn's post hoc (Continued on next page)

Fig 1 (Continued)

test). (c) Alpha diversities of microbial communities among the three age groups (*P* value was calculated using a Kruskal–Wallis test and Dunn's post hoc test). (d) PERMANOVA (left) and PCoA analysis (middle: age, right: sex), using Bray–Curtis distances. (e) PERMANOVA (left) and PCoA analysis (middle: age, right: sex), using Jaccard distances. *$P < 0.05$, **$P < 0.01$, ***$P < 0.001$.

data, where the presence of a microbe is represented as 1 and absence as 0, provides evidence for the feasibility of using binarized data for age differentiation and prediction.

## Signal-to-noise ratio (SNR) assessment of four data processing methods

To further evaluate which data processing approach is most suitable for microbiome-based age inference, we introduced the concept of the signal-to-noise ratio (SNR), which

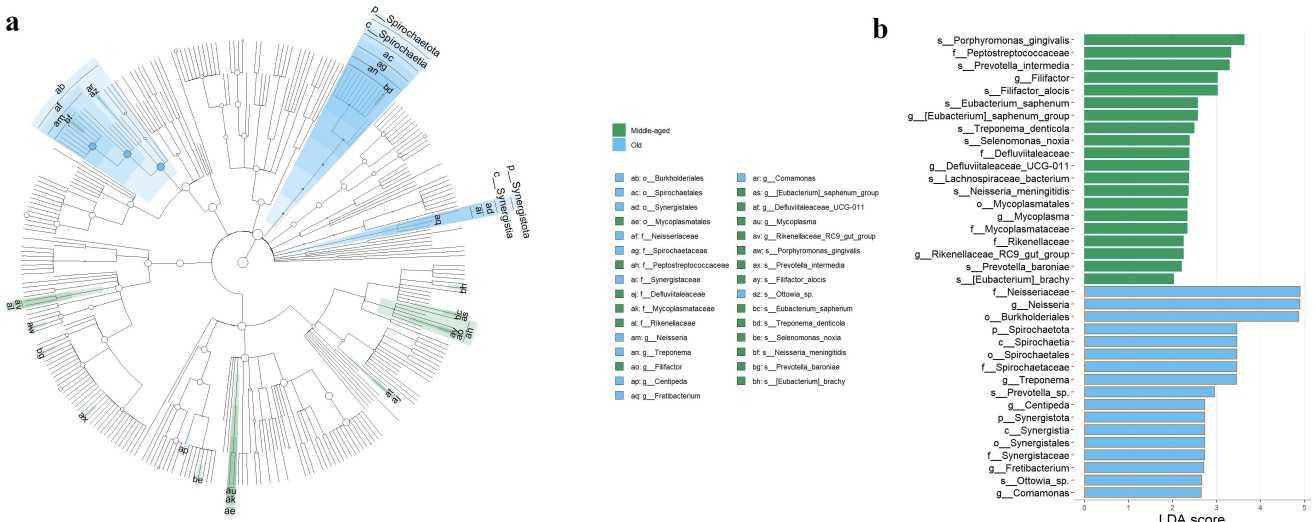

**c** (Relative abundance differences in the Middle-aged group at genus level)

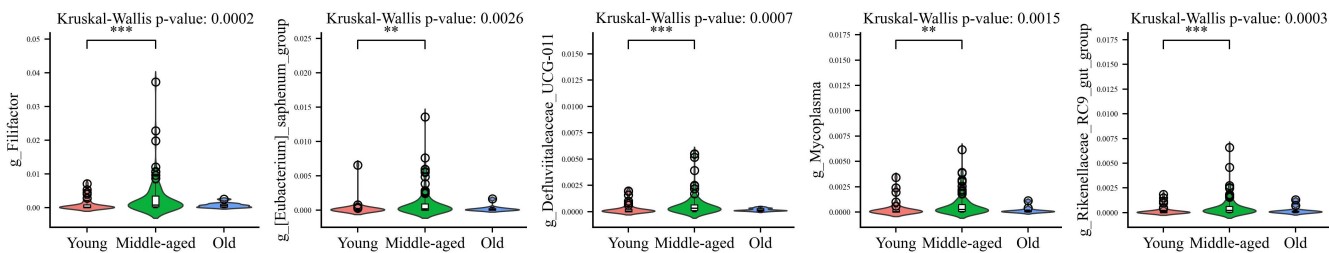

**d** (Relative abundance differences in the Old group at genus level)

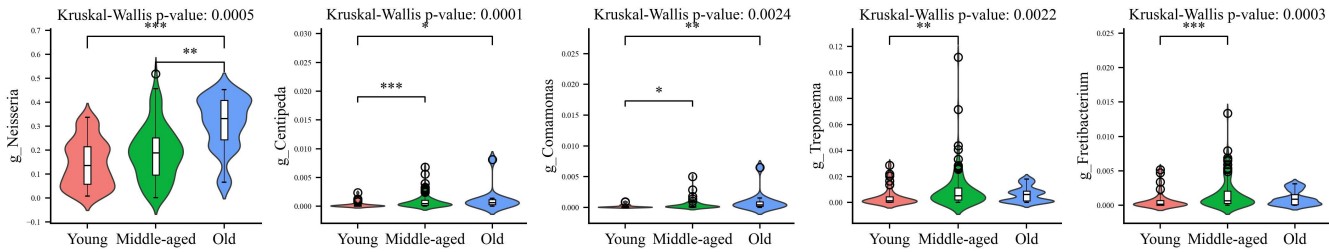

**FIG 2** (a, b) The LEfSe cladogram (LDA > 2.0, *P* < 0.05) shows differentially abundant taxa among three groups (middle-aged group: green; old group: blue; no differentially abundant taxa in the young group). (c, d) Violin plots of relative abundance of bacterial taxa (genus level) identified as significantly contributing to differences in the LEfSe analysis. (c) Relatively higher abundance in the middle-aged group. (d) Relatively higher abundance in the old group. *P* value was calculated using the Kruskal–Wallis test followed by Dunn's post hoc test. *$P < 0.05$, **$P < 0.01$, ***$P < 0.001$.

compares inter-group variation (signal) to intra-group variation (noise) (see Materials and methods for details). Four data processing strategies were considered: binarization, relative abundance, log2 transformation, and centered log-ratio (CLR) transformation.

A total of 46,263 ASVs were obtained from sequencing 150 samples. Each ASV was transformed using the four methods, and SNR values were calculated based on age groupings (young: 0–29 years; middle-aged: 30–64 years; old: ≥65 years). Three ASVs (ASV_4788, ASV_12315, and ASV_18698) were detected in all 150 samples and thus exhibited no inter- or intra-group variability in the binarized data, rendering their SNR values undefined. For the remaining 46,260 ASVs, the average SNRs were as follows: 1.1420 for binarized data, 1.1243 for relative abundance, 1.1435 for log2-transformed data, and 1.1479 for CLR-transformed data (Table S3-1).

Wilcoxon signed-rank tests were performed to compare the SNR values between all transformation methods. All comparisons yielded statistically significant results as determined by the Wilcoxon signed-rank test ($P < 0.05$, Table S3-2), which was chosen due to the non-normal distribution of the data. These findings suggested that binarization, log2, and CLR transformations provided superior discriminative power for age prediction compared to the conventional relative abundance approach, with CLR performing best overall.

Furthermore, pairwise comparisons were conducted for each ASV across all processing methods. The binarized approach yielded a greater number of ASVs with higher SNRs when compared to each of the other three methods (Table S3-3). Similarly, when counting the number of ASVs with SNR > 1, the binarized method again yielded the highest count (14,938 ASVs), followed closely by CLR (14,929), log2-transformed (14,929), and relative abundance data (14,550) (Table S3-4).

## Feature selection for age association via LASSO

To prevent potential overfitting during subsequent model construction, the 150 samples were randomly divided into a training set ($n = 135$) and an independent validation set ($n = 15$) at a ratio of 9:1 (detailed in Materials and methods). Age-associated ASV feature selection was performed exclusively within the training set. A total of 46,263 ASVs from the 135 samples were included in the analysis. Least Absolute Shrinkage and Selection Operator (LASSO) regression was employed to identify ASVs with potential predictive value. To ensure the robustness and stability of feature selection, 100 iterations of LASSO were conducted using random seeds ranging from 1 to 100. A threshold of 0.5 was applied, such that ASVs appearing in more than 50 iterations were retained.

As a result, 89 ASVs were selected from binarized data, 121 from CLR-transformed data, 118 from log2-transformed data, and 26 from relative abundance data (Table S4-1,2,3,4). These selected features were used for subsequent model tuning and construction. Although the sets of age-associated ASVs varied across the four data processing strategies, the majority of selected ASVs were consistently affiliated with the phyla *Firmicutes* and *Proteobacteria* (binarized data: 26/89 *Firmicutes*, 15/89 *Proteobacteria*; CLR-transformed data: 31/121 *Firmicutes*, 21/121 *Proteobacteria*; log2-transformed data: 30/118 *Firmicutes*, 22/118 *Proteobacteria*; and relative abundance data: 9/26 *Firmicutes*, 6/26 *Proteobacteria*). Notably, 14 ASVs were commonly identified across all four data types, among which 57.14% (8/14) were from *Firmicutes* and *Proteobacteria*. This taxonomic distribution aligns with findings from the differential abundance analysis based on relative abundance data, where most significant taxa also belonged to these two phyla.

Among all pairwise comparisons, the highest overlap in selected ASVs was observed between log2- and CLR-transformed data, with 116 shared ASVs. This is likely attributable to the fact that both transformations are log-based and therefore impose similar effects on data scale and distribution. Interestingly, binarized data yielded the largest number of unique ASVs ($n = 17$), compared to 4, 2, and 8 unique ASVs for CLR-transformed, log2-transformed, and relative abundance data, respectively (Fig. 3a). This may suggest

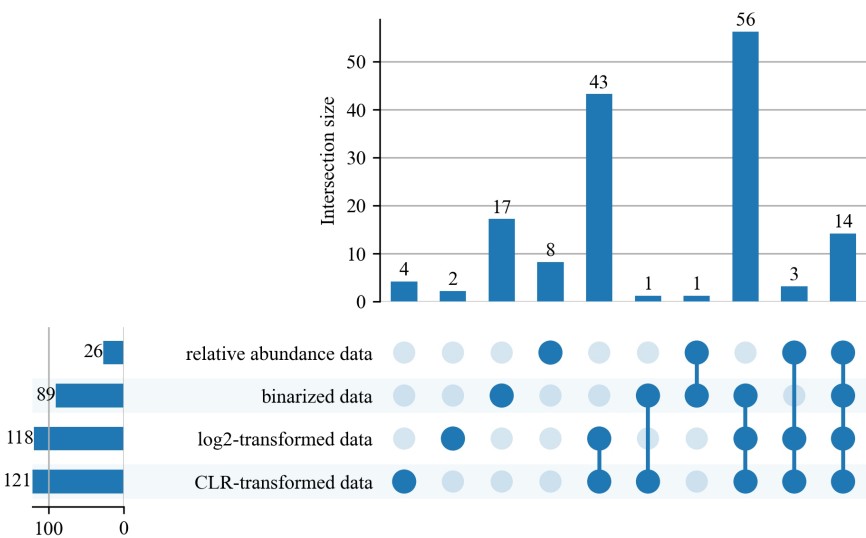

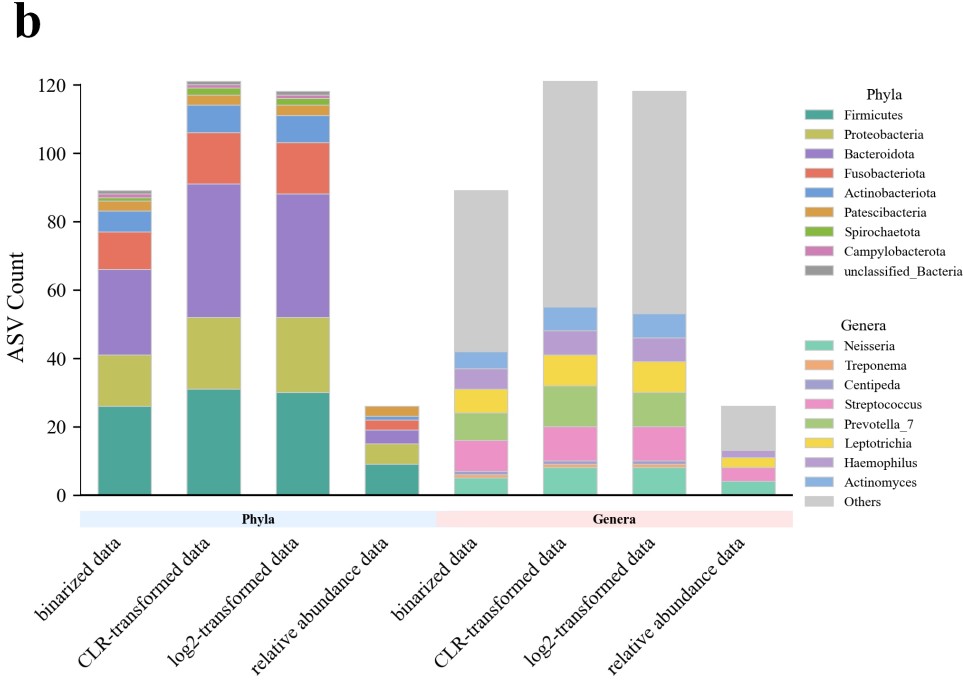

**FIG 3** Comparison of ASVs selected by LASSO across four microbial data processing methods. (a) UpSet plot showing the overlap of selected ASVs. (b) Taxonomic annotation of selected ASVs at the phylum and genus levels.

that binarization captures unique microbial signatures potentially informative for age prediction.

When comparing the LASSO-selected ASVs to taxa identified via LEfSe, only *Neisseria* was consistently detected at the genus level across all four data types (binarized: 5 ASVs; CLR: 8 ASVs; log2: 8 ASVs; relative abundance: 4 ASVs). Additionally, *Treponema* and *Centipeda* were identified in the binarized, CLR-transformed, and log2-transformed datasets, but each contained only a single ASV (Fig. 3b). This discrepancy may stem from the fact that LEfSe relies on predefined categorical groupings, which may not be optimal for identifying features with continuous or linear relationships to variables such as age

(20). In contrast, LASSO-based selection assumes a potential linear association with age, offering a more suitable framework for this context.

## Classifier-based ensemble model: mechanism and hyperparameter tuning

To improve the accuracy of individual age prediction based on oral microbial data, we developed a model using age bins (i.e., dividing the age data into different age intervals) as output combined with a voting method (21). Given the advantages of multiple classifier combination methodologies in enhancing predictive accuracy (17), the ensemble model was constructed by integrating multiple classifiers (Fig. 4), and the number of classifiers depends on the width of age bin. Assuming the width of the age bin is *N* years and the age range spans from 1 to 80 years, the number of classifiers will vary based on *N*.

When *N* = 20, there will be 20 classifiers:

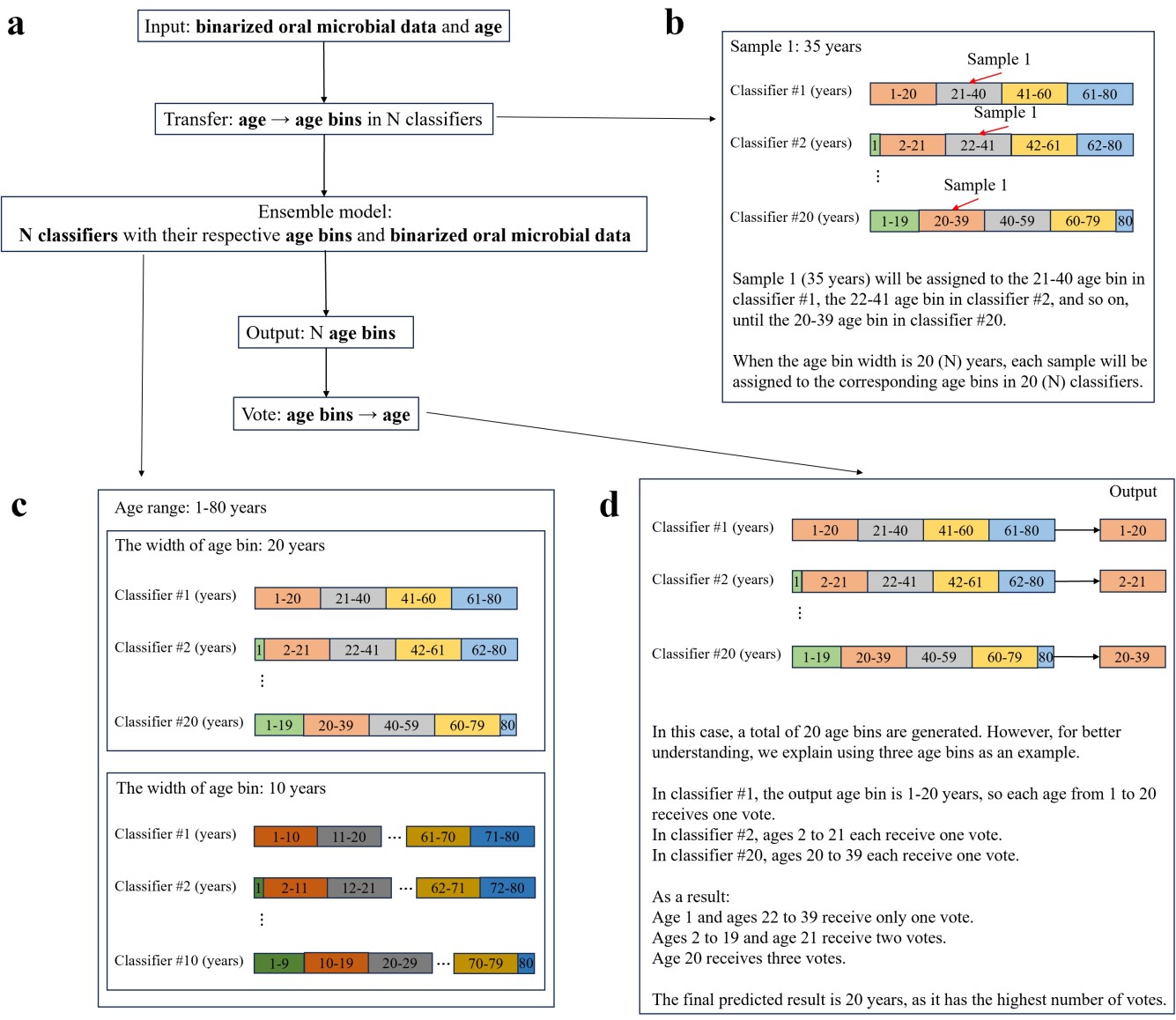

**FIG 4** The principle of ensemble model with N classifiers. (a) Operation process of the ensemble model. (b) The transformation of sample age into age bins under specific classifiers during training. (c) The illustration of the classifiers in the ensemble model for age prediction. (d) The process of voting to derive the final predicted age.

Classifier #1: 1–20, 21–40, 41–60, 61–80
Classifier #2: 1, 2–21, 22–41, 42–61, 62–80
Subsequent classifiers shift the bins by 1 year until classifier #20: 1–19, 20–39, 40–59, 60–79, 80

When $N = 10$, there will be 10 classifiers:

Classifier #1: 1–10, 11–20, 21–30, …, 71–80
Classifier #2: 1, 2–11, 12–21, …, 72–80
Until classifier #10: 1–9, 10–19, 20–29, …, 70–79, 80 (Fig. 4C)

During the model training process, all classifiers are trained on the same dataset. A given sample will be assigned to a specific age bin within each classifier in the ensemble. For example, assuming a sample age of 35 years, when $N = 20$, it falls into the age bin of 21–40 years in classifier #1 and the 22–41 bin in classifier #2 (Fig. 4b). When $N = 10$, it is assigned to the 31–40 bin in classifier #1, the 32–41 bin in classifier #2, and so forth. The model is ultimately trained using the subjects' oral microbial data as input and the corresponding age bin as output. During prediction, the oral microbial data are input into the model, and then the ensemble model produces an age bin prediction. Each age within the predicted age bin is assigned one vote. After all classifiers make their predictions, the votes are aggregated, and the age with the highest vote count is designated as the predicted age. For example, consider a sample under the condition that $N = 20$. In classifier #1, the output age bin is 1–20; in classifier #2, the output age bin is 2–21 years; and in classifier #3, the output age bin is 20–39 years. In reality, there will be 20 classifiers as a result, but here, for ease of explanation, we consider that there are only 3 classifiers. The result of the voting is that age 20 receives 3 votes; ages 2–19 and 21 receive 2 votes; ages 1 and 22–39 receive 1 vote; and the final predicted age is 20 years. In cases where multiple ages receive the highest number of votes, the youngest age is prioritized (Fig. 4d).

Given that the participants' ages in this study ranged from 6 to 78 years, we set the age range between 1 and 80 years. Considering the substantial computational time required to tune hyperparameters for all possible bin widths, we initially fixed the age bin width at 20 years. To construct the ensemble models, we evaluated eight commonly used machine learning algorithms: K-nearest neighbors (KNN), linear discriminant analysis (LDA), logistic regression (LR), Naïve Bayes, neural network, RF, SVM, and XGBoost. To minimize algorithm-specific biases and maximize predictive performance, we conducted hyperparameter optimization based on 135 training samples for both the ensemble models and five traditional models (KNN, neural network, RF, support vector regression/SVR, and XGBoost) used for comparison. Detailed hyperparameter settings are provided in the Materials and methods section. A total of four types of microbial data were used for model construction and participated in hyperparameter optimization, including binarized data, CLR-transformed data, log2-transformed data, and relative abundance data. Hyperparameters were tuned using grid search with tenfold cross-validation on the training set ($n = 135$). The optimal hyperparameter settings are summarized in Table S5-1 and 2. Detailed tuning procedures, along with the corresponding code and results, are available at https://github.com/Z-yuxiang00/Oral_Microbiome-Age/tree/main/4_Hyperparameter_tuning.

## Age prediction from oral microbiome

In the ensemble model we constructed, three main factors influenced the accuracy of age prediction: the algorithm, the width of the age bin, and the defined age range. To identify the optimal predictive model, we first defined the age range as 1 to 80 years based on the sample age distribution (6–78 years). Subsequently, we selected eight commonly used classification models and performed age prediction for each model under different age bin widths (1–64 years). To prevent overfitting during the

age bin width selection process, we performed tenfold cross-validation on the 135 training samples. For each machine learning algorithm, 64 MAE values were obtained, corresponding to age bin widths ranging from 1 to 64 years. Within each algorithm, the age bin width yielding the lowest MAE on the training set was selected as the optimal bin width. A comprehensive comparison of the 64 training MAE values across all eight machine learning algorithms and four microbial data types is shown in Fig. S2 and Table S6. The optimal training MAE values, their corresponding age bin widths, and the independent validation results for each machine learning algorithm and data type are summarized in Table S5-1. These models were subsequently evaluated on the independent validation set ($n = 15$), and the one achieving the lowest MAE was selected as the final model for downstream application.

The results revealed that the top four performing combinations were as follows: binarized data with XGBoost (MAE = 7.20 years, $N = 32$), log2-transformed data with LDA (MAE = 7.33 years, $N = 35$), log2-transformed data with KNN (MAE = 7.80 years, $N = 15$), and relative abundance data with XGBoost (MAE = 7.80 years, $N = 43$) (Table 1; Table S5-1). Given that ensemble models may underperform at the extreme ends of the age distribution due to inherent model limitations, we additionally evaluated MAE within the 20–59-year age range ($n = 12$). In this subset, the MAEs were 4.33 years for binarized data combined with XGBoost, 5.42 years for log2-transformed data combined with LDA, 5.00 years for log2-transformed data combined with KNN, and 5.92 years for relative abundance data with XGBoost (Table 2). Considering these results in conjunction with the SNR-based and feature selection analyses, we propose that the combination of binarized data and XGBoost represents the most promising configuration and has therefore adopted it for downstream analyses.

Considering current mainstream approaches for microbiome-based age prediction primarily rely on regression models using relative abundance data (3, 20), we further evaluated whether our ensemble modeling framework could improve predictive accuracy compared to traditional regression-based methods. After excluding Naïve Bayes, LDA, and logistic regression, which are not inherently suitable for regression analysis, we constructed regression-based age prediction models using the remaining five machine learning algorithms. In the binarized data, ensemble models outperformed traditional regression models across all five comparable algorithms except for RF. A similar pattern was observed in the CLR-transformed data, where ensemble models again outperformed their traditional counterparts in all applicable cases except RF. For the log2-transformed data, ensemble models built with KNN and RF demonstrated superior performance, whereas the performance of ensemble and traditional models was comparable for the other three algorithms. In the relative abundance data, ensemble models outperformed traditional models in neural network and XGBoost, while traditional models yielded better results for the remaining three algorithms (Table 1). Overall, ensemble models showed a performance advantage in most scenarios, particularly when applied to binarized and CLR-transformed data. These findings suggest

**TABLE 1** Comparison of MAE on the independent validation set across four data processing methods in all age groups

| Algorithm | MAE value of binarized data | | MAE value of CLR-transformed data | | MAE value of log2-transformed data | | MAE value of relative abundance data | |
|---|---|---|---|---|---|---|---|---|
| | Ensemble | Traditional | Ensemble | Traditional | Ensemble | Traditional | Ensemble | Traditional |
| KNN | 9.13 | 9.41 | 9.27 | 9.38 | 7.80 | 9.2 | 13.47 | 12.43 |
| LDA | 10.67 | –[a] | 8.53 | – | 7.33 | – | 10.13 | – |
| LR | 9.27 | – | 8.60 | – | 8.20 | – | 12.13 | – |
| Naïve Bayes | 9.20 | – | 8.53 | – | 9.40 | – | 9.67 | – |
| Neural network | 9.87 | 9.94 | 8.47 | 9.70 | 8.60 | 8.56 | 8.20 | 9.25 |
| RF | 9.73 | 9.31 | 10.47 | 9.64 | 8.40 | 8.61 | 9.00 | 8.82 |
| SVM | 8.87 | 9.46 | 8.00 | 8.48 | 9.07 | 9.06 | 12.00 | 11.80 |
| XGBoost | 7.20 | 8.11 | 8.20 | 9.57 | 9.40 | 8.28 | 7.80 | 7.86 |

[a]"–" indicates that no data are available for that entry.

TABLE 2  Comparison of MAE on the independent validation set across four data processing methods in the 20–59-year age group based on the best-performing models in all-age evaluation

| Algorithm | MAE value of binarized data | | MAE value of CLR-transformed data | | MAE value of log2-transformed data | | MAE value of relative abundance data | |
|---|---|---|---|---|---|---|---|---|
| | Ensemble | Traditional | Ensemble | Traditional | Ensemble | Traditional | Ensemble | Traditional |
| KNN | 6.33 | 6.46 | 5.92 | 6.16 | 5.00 | 6.01 | 12.33 | 10.72 |
| LDA | 8.50 | –[a] | 6.83 | – | 5.42 | – | 9.58 | – |
| LR | 6.67 | – | 4.58 | – | 5.17 | – | 9.75 | – |
| Naïve Bayes | 6.92 | – | 6.17 | – | 7.17 | – | 9.58 | – |
| Neural network | 6.75 | 7.09 | 5.50 | 7.14 | 5.75 | 5.85 | 8.33 | 8.63 |
| RF | 7.25 | 6.97 | 7.00 | 6.92 | 6.00 | 6.27 | 7.00 | 7.00 |
| SVM | 6.42 | 6.71 | 5.08 | 5.84 | 6.00 | 6.15 | 10.50 | 10.4 |
| XGBoost | 4.33 | 5.60 | 5.92 | 7.16 | 7.08 | 5.99 | 5.92 | 6.64 |

[a]"–" indicates that no data are available for that entry.

that the ensemble model, particularly when combined with binarized data and the XGBoost algorithm, represents one of the most effective strategies currently available for microbiome-based age prediction.

Regarding the selection of the age range, it is crucial to ensure that it encompasses all target samples. Whether to extend the range appropriately depends on the characteristics of the prediction samples. For instance, in a forensic case, if the age of the suspect is estimated to be within the range of 80–90 years (as an extreme case), the age range needs to be extended to 90 years or even 100 years. To investigate whether expanding the age range affects prediction accuracy, we employed an XGBoost algorithm, using an age bin of 32 to construct an ensemble model. Different age ranges were tested, including 6–78 years, 1–80 years, 1–90 years, and 1–100 years. The results revealed that the MAE was 8.00 years for the age range of 6–78 years. When the age range was expanded to 1–80 years, 1–90 years, and 1–100 years, the MAE improved to 7.20 years (Table S7). Our findings indicate that appropriately broadening the age range enhances prediction accuracy in this dataset.

## Application of the ensemble model on a larger dataset, comparison with the existing model, and evaluation of the impact of the sex dummy variable on prediction accuracy

To further evaluate the model, we applied the combination of binarized data, ensemble model, and XGBoost to a larger dataset consisting of 2,550 samples, which were collected from the United States, the United Kingdom, Tanzania, and other countries (3). The sequencing data consisted of 16 S-V4 rRNA gene amplicon data, with an age range from 18 to 85 years. The dataset was split into a training set (*n* = 2295) and an independent validation set (*n* = 255) using a 9:1 ratio. After binarizing the dataset, it was incorporated into the ensemble model, with the age range set to 1–90 years to accommodate the actual range of 18–85 years. The results showed that the MAE for the independent validation set was 4.80 years, with the MAE for the 20–59 age group (*n* = 232) reaching 3.76 years (Fig. 5a).

Furthermore, we compared our ensemble model with a previously published age prediction model for the oral microbiome (3). According to the authors' code, this model employed fivefold cross-validation on the 2,550 samples for age prediction. To ensure consistency, we adopted the same cross-validation strategy with the random seed set to 8. After binarizing the data, LASSO analysis was conducted on the training set in each fold to identify age-associated ASVs. The binarized training and testing sets were then used as inputs to our ensemble model, and the MAE was computed for the testing set in each fold. Results from the fivefold cross-validation showed that our ensemble model achieved an MAE of 4.44 ± 0.34 years, outperforming the previously reported model (MAE = 4.94 ± 0.23 years) (Fig. 5b). To further evaluate whether this difference was statistically significant, we aggregated predictions across all folds and calculated the

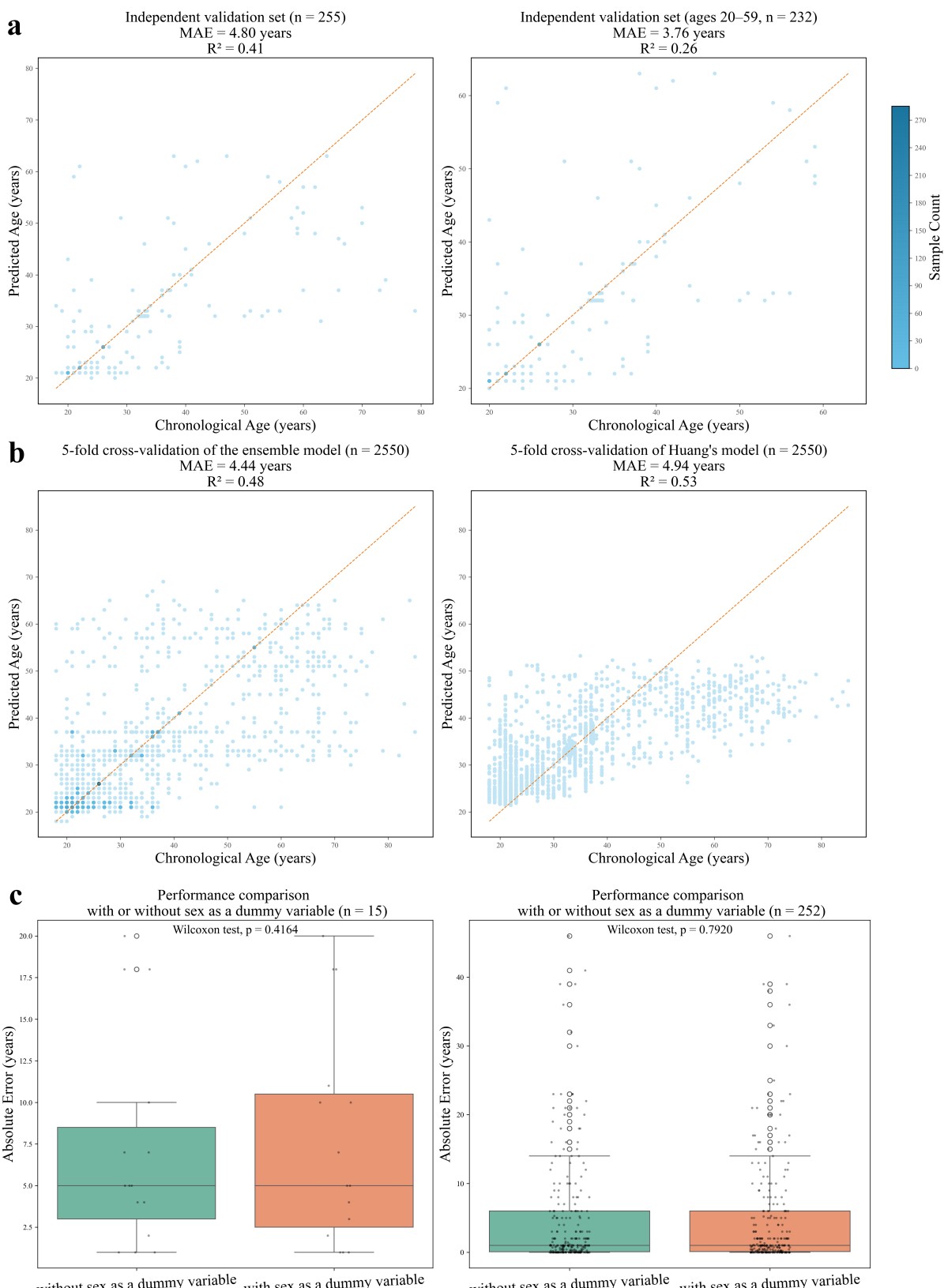

**FIG 5** (a) Age prediction performance on the independent validation sets of 2,550 samples. Left: full validation set (*n* = 255). Right: age 20–59 subset (*n* = 232). (b) Fivefold cross-validation performance on the full dataset (*n* = 2550) using the ensemble model (left) and Huang et al.'s model (right). (c) Effect of incorporating sex as a dummy variable on prediction accuracy. Left: the independent validation set (*n* = 15) from the 150-sample cohort. Right: the independent validation set (*n* = 252) from the 2,516-sample cohort. The Wilcoxon signed-rank test was used to evaluate statistical differences.

absolute error (i.e., the absolute difference between predicted and actual ages) for each individual sample. A pairwise comparison between the two models was then conducted using the Wilcoxon signed-rank test, due to the non-normal distribution of the data. The test yielded a *P*-value < 0.05, indicating a statistically significant improvement in performance by the ensemble model using binarized oral data compared to the previous method (Table S8-1 and 2).

Considering that sex was identified as another significant covariate influencing microbial composition in PERMANOVA analysis, we further assessed whether sex affected model performance. To this end, we stratified both the 150-sample dataset in this study and the 2,516-sample dataset from Huang et al. (originally 2,550 samples, with 34 excluded due to missing sex information) by sex, resulting in two subsets per cohort. We first trained the model using male samples and evaluated its performance on female samples and then reversed the process. In the 150-sample dataset, the MAE was 12.77 years when male samples (*n* = 73) were used as the validation set, and 12.68 years when female samples (*n* = 77) were used. In the 2516-sample dataset, the MAE was 6.42 years for male validation samples (*n* = 1342) and 7.44 years for female validation samples (*n* = 1174) (Table S9-1). Across both datasets, the MAE differences between male and female samples were minimal, consistent with the findings reported by Huang et al. (3).

To further evaluate the potential confounding effect of sex, we introduced a dummy variable (female = 0, male = 1) (22) into the model as an additional feature, allowing the model to adjust for sex. In the 150-sample dataset (9:1 train-test split), the MAE of the validation set (*n* = 15) slightly increased from 7.20 years to 7.73 years after incorporating the sex variable (Table S9-2). However, a Wilcoxon signed-rank test revealed no significant difference (*P* = 0.4164, Fig. 5C; Table S9-3). We then excluded the 34 samples with missing sex information from the 2,550-sample dataset, removing 31 from the training set (reducing it from 2,295 to 2,264) and 3 from the testing set (reducing it from 255 to 252). Model performance with and without the sex dummy variable was evaluated. The MAE of the validation set was 4.77 years without the sex variable and 4.76 years with it. Again, the Wilcoxon signed-rank test indicated no statistically significant difference (*P* = 0.7920, Fig. 5C; Table S9-3). Together, these results suggest that sex has a limited impact on the model's age prediction performance in our study.

## DISCUSSION

In this study, we demonstrated that host age can be accurately predicted based on oral microbial ASV profiles, highlighting the potential of oral microbiome-based approaches in various applications. One of the key advantages of this method is its non-invasive nature, allowing age estimation using easily accessible biological samples such as saliva. This provides a promising alternative to traditional age determination techniques that often require more invasive procedures. Moreover, by comparing microbiome-predicted age with chronological age, this approach may serve as a personalized health assessment tool, offering insights into deviations that could be linked to accelerated aging or underlying health conditions. In addition to its biomedical implications, microbial age prediction also holds forensic relevance, as it may aid in age estimation for unidentified individuals in forensic investigations, thereby contributing to legal and criminal casework.

In terms of microbial composition analysis, the predominant bacterial phyla identified in the 150 Chinese oral saliva samples included *Proteobacteria*, *Firmicutes*, and *Bacteroidota*. Although certain bacterial taxa exhibited differences, they were generally comparable to the major bacterial taxa reported in previous studies on populations from the United Kingdom, Finland, Qatar, and South Korea (6, 23–25). A similar pattern was also observed at the genus level. Furthermore, the Kruskal–Wallis test revealed that the relative abundance of *Proteobacteria* increased with age, a trend previously reported in the small bowel microbiome (26). In contrast, the phylum *Firmicutes* exhibited an age-related decline, a pattern documented in the fecal microbiome (27). The consistent

age-related trends of specific microbial taxa across different sample types suggest the potential for developing cross-sample-type age prediction models in future studies.

Furthermore, changes in the abundance of *Proteobacteria* and *Firmicutes* during aging may be closely related to host immune metabolism. Studies have shown that aging is positively correlated with increased systemic inflammation (26). In addition, Biagi et al. (28) found that the enrichment of *Proteobacteria* in the gut was significantly positively correlated with the increase of pro-inflammatory cytokines IL-6 and IL-8. Moreover, Leite et al. (26) suggested that the increase in *Proteobacteria* could be associated with age-related shifts in microbial metabolic pathways, particularly those related to the ubiquinone biosynthesis pathway. The ubiquinone biosynthesis pathway, an antioxidant pathway, is upregulated under stress conditions such as hypoxia. As aging progresses, the degree of hypoxia in the small intestine increases, and microbial genes associated with the ubiquinone biosynthesis pathway are also enriched in the duodenal microbiome of the elderly. In contrast, for *Firmicutes*, which decrease with age, many members of this phylum are key butyrate producers. Given butyrate's anti-inflammatory properties (29) and its reported role in extending lifespan in model organisms (30), the depletion of this phylum may further exacerbate chronic inflammation in aging hosts, contributing to systemic inflammation (26, 31).

The α-diversity analysis based on observed species richness and the Chao1 index revealed that microbial diversity increased with age in this study, which contrasts with some previous findings (6, 32). This discrepancy may stem from the fact that the primary factor driving the decline in microbial diversity is oral disease rather than aging itself. That is, as age increases, the incidence of oral diseases rises, which in turn leads to reduced microbial diversity. This explanation is supported by the study conducted by Murugesan et al., which demonstrated that abnormal oral conditions are associated with lower microbial diversity and richness (6). A similar pattern has also been observed in gut microbiota (20, 33). In this study, we primarily selected individuals with good oral health as our study population. Therefore, when investigating the relationship between age and oral microbiome diversity, the health status of participants should be carefully considered.

The PERMANOVA analyses based on Bray–Curtis and Jaccard distances both indicated that age grouping is one of the primary factors shaping microbial composition. Notably, the PERMANOVA analysis based on Jaccard distances suggests that binarized oral microbial data holds promise as a potential predictor for age inference. Subsequent signal-to-noise ratio analyses for four data types (binarized, CLR-transformed, log2-transformed, and relative abundance data), along with further analysis of the ASV features selected by LASSO, have further confirmed this finding. A key advantage of binarized data lies in its discretization process, which transforms quantitative data into qualitative categories (11). Discretization can be regarded as a data reduction technique, as it maps a vast range of numerical values into a significantly reduced set of discrete categories (34). This approach offers several benefits: it simplifies and reduces the dataset, leading to faster learning speeds and more compact, interpretable results; it mitigates potential noise in the data, effectively concealing imperfections in the original dataset; and it enhances the interpretability and usability of the data for researchers and practitioners (11, 34, 35). Moreover, the previous literature explicitly states that, from a statistical perspective, binary covariates may be preferable as they (1) facilitate a clearer interpretation of common effect measures from statistical models, such as odds ratios and relative risks; (2) circumvent the linearity assumption inherent in statistical models for continuous covariates; (3) allow for modeling of previously suspected or predefined threshold effects; and (4) enhance the efficiency of data summarization (36). Furthermore, age binning in ensemble models is also a form of discretization, which helps reduce the granularity of information systems, improve the performance and learning accuracy of data mining and machine learning algorithms, and enhance classification, clustering, and noise resistance capabilities (11, 18).

Furthermore, the PERMANOVA analysis also indicated that sex is a factor influencing microbial composition differences between groups. Therefore, after selecting binarized data and combining it with an ensemble model of 32 XGBoost classifiers, we performed two steps: (1) dummy variable processing and (2) splitting the dataset by sex for cross-validation (i.e., using the male subset as the training set and the female subset as the testing set and vice versa). The results revealed that, regardless of whether dummy variables or sex-based subsets were used for training and validation, there was no significant deviation in the overall outcomes. Hence, although the PERMANOVA analysis suggested a potential impact of sex on the microbial composition across different groups, the influence of sex on model construction was found to be relatively weak. This may be attributed to the fact that the PERMANOVA analysis was based on predefined groupings, while the models were constructed based on continuous variables.

Regarding ensemble model performance, a necessary and sufficient condition for an ensemble of classifiers to outperform its individual members is that the classifiers must be both accurate and diverse (17, 18, 37). Several strategies for constructing ensemble models have been summarized in the literature, including manipulating the training examples, input features, and output targets, as well as injecting randomness (17). In this study, we focused on output target manipulation by partitioning the output variable (age) into different bins across classifiers, ensuring that each classifier was trained on a unique binning scheme. This approach maximizes classifier independence while covering the full age spectrum under a fixed bin width. Moreover, well-established machine learning algorithms, such as neural networks, decision trees, and bagging, have been demonstrated to enhance ensemble model performance, providing reliable accuracy (16, 37). Additionally, findings by Ahmad et al. suggest that ensemble methods are effective for regression tasks (18). Thus, the ensemble model employed in this study is expected to outperform individual models, offering more accurate and stable predictions.

The varying performance observed across algorithms in this study can be attributed to fundamental differences in their underlying assumptions, flexibility, and suitability for modeling high-dimensional, sparse, and over-dispersed microbiome data (9, 38). Among these, the XGBoost algorithm demonstrated superior predictive power. This advantage may be attributable to a series of systems and algorithmic optimizations, including a sparsity-aware tree learning framework, regularization techniques to mitigate overfitting, feature selection capability, and highly scalable parallel and out-of-core computation (39, 40), which together enable it to efficiently capture complex patterns inherent in microbiome data. Previous studies (41, 42) have also consistently demonstrated the superiority of this approach in terms of accuracy and generalizability across regression and classification tasks.

Algorithms such as LR, LDA, and Naïve Bayes remain widely used in high-dimensional data due to their simplicity, interpretability, and computational efficiency. For instance, LR estimates a set of weights to model class probabilities as a linear function of input features. While this method performed adequately in many settings, the performance might be affected when sample sizes are limited, particularly when fewer than 100 samples are available per class (38, 43). In contrast, KNN and SVM offer non-linear modeling capabilities with greater flexibility. KNN is a non-parametric method that classifies samples based on local neighborhood voting in feature space (44). SVM, equipped with kernel functions and maximal margin hyperplanes, performs well in settings with high variable-to-sample ratios and includes regularization strategies to mitigate overfitting (45, 46). However, the computational cost of the method may become a limiting factor when the number of features is extremely large (38). Other advanced machine learning algorithms, such as RF and NN, have gained attention for their ability to model complex relationships in biological data (15, 38, 43). RF models often achieve competitive performance and are relatively robust to overfitting, especially when appropriate constraints such as maximum tree depth are applied. However, they may be sensitive to noise if not properly tuned (38). Neural networks are capable of

capturing intricate patterns and have shown promising results across omics applications, although their interpretability remains limited compared to simpler models, which may pose challenges for extracting biological insights (15).

Although the overall results are robust, this study has certain limitations. First, while five potential confounding factors were recorded, the number of participants who smoked or consumed alcohol was relatively small (Table S10-1). This was due to our study design, which aimed to minimize additional confounding variables to better isolate the effects of age on the microbiome. As a result, other unaccounted factors that may significantly shape microbial composition, such as geographic location, environmental exposures, and disease status (6, 24), could not be fully assessed. Second, the ensemble model exhibited suboptimal performance in predicting ages at both the youngest and oldest extremes, likely due to the relatively small sample sizes in these age groups or potential edge effects introduced by discretizing age. Increasing the number of samples in these age ranges or developing an alternative approach (such as oversampling or weighted loss functions) may help mitigate these limitations. Third, the use of binarized data introduces a potential limitation: if a given ASV is present in all samples under investigation, binarization eliminates its discriminatory power, as the feature becomes invariant (i.e., all values equal to 1). This issue was observed with ASV_4788, ASV_12315, and ASV_18698 in our study. When a large number of such universally present ASVs exist, we recommend using CLR- or log2-transformed data, or alternatively, relying on conventional relative abundance data for analysis.

In summary, we comprehensively investigated the relationship between age and the oral microbiome, identifying age as a key factor shaping microbial community composition. Building on this, we introduced an innovative approach by leveraging binarized microbial ASV data as predictive features for age estimation and developed an ensemble model based on the XGBoost algorithm, comprising 32 classifiers. This model achieved impressive prediction accuracy, with an MAE of 7.20 years in the independent validation set ($n = 15$), and 4.33 years in the 20–59 age subgroup ($n = 12$), significantly outperforming traditional machine learning models. When the sample size increased to 2,550, the MAE in the independent validation set ($n = 255$) was reduced to 4.80 years, with the 20–59 age subgroup ($n = 232$) achieving an MAE of 3.76 years, underscoring the model's generalizability and robustness. Moreover, when compared to the previously published model for age prediction based on oral microbiomes, our model demonstrated significantly superior performance. Taken together, these findings highlight the potential of binarized oral microbial data, in conjunction with our ensemble model, for accurate age prediction, laying a robust foundation for future age-related research.

## MATERIALS AND METHODS

### Sample collection and DNA extraction

A total of 150 volunteers participated in this study, ranging in age from 6 to 78 years. All participants self-reported no antibiotic usage in the three months prior to enrollment. After the purpose and procedures of the study were explained to all donors, written informed consent was obtained from them or their legal guardians. Data on participants' age, BMI, sex, smoking status, and alcohol consumption were collected (Table S10-1). The study was approved by the Ethical Committee of Sichuan University (K2020033). The 2,550 oral samples were collected from previous studies(3) (Table S10-2).

All participants were instructed to abstain from food and drink for at least 2 h prior to saliva collection. Immediately after collection, saliva samples were stored at −20°C. Once all samples were collected, they were transported to the laboratory under frozen conditions for further processing. Genomic DNA was extracted using the QIAamp DNA Mini Kit (QIAGEN, Hilden, Germany) following the manufacturer's protocol. DNA concentrations were determined using a NanoDrop 2000 spectrophotometer (Thermo Fisher Scientific, Waltham, USA).

## 16S rRNA gene amplicon sequencing and sequence analysis

The V3–V4 hypervariable regions of the 16S rRNA gene were amplified using the forward primer 341F (5′-CCTAYGGGRBGCASCAG-3′) and the reverse primer 806R (5′-GGAC-TACNNGGGTATCTAAT-3′). Sample-specific 7 bp barcodes were incorporated into the primers to facilitate multiplex sequencing. The 25 µL PCR reaction mixture consisted of 5 µL of reaction buffer (5×), 1 µL of each forward and reverse primer (10 µM), 2 µL of dNTPs (2.5 mM), 1 µL of DNA template, 14.75 µL of nuclease-free water, and 0.25 µL of Fast pfu DNA polymerase (5 U/µL). PCR thermal cycling conditions included an initial denaturation at 98℃ for 5 min, followed by 25 cycles of denaturation at 98℃ for 30 s, annealing at 53℃ for 30 s, and extension at 72℃ for 45 s, with a final elongation step at 72℃ for 5 min.

To eliminate residual contaminants and PCR artifacts, amplicons were purified using Vazyme VAHTS DNA Clean Beads (Vazyme, Nanjing, China), following the manufacturer's protocol. The quality and concentration of purified amplicons were assessed using the Quant-iT PicoGreen dsDNA Assay Kit (Invitrogen, Carlsbad, CA, USA). Following individual quantification, equimolar amounts of purified amplicons were pooled and subjected to paired-end 2 × 250 bp sequencing on the Illumina NovaSeq platform, utilizing the NovaSeq 6000 SP Reagent Kit (500 cycles). Sequencing was performed at Shanghai Personal Biotechnology Co., Ltd. (Shanghai, China).

Microbiome bioinformatics analyses were performed using QIIME2 (v2022.11) with slight modifications based on the official tutorials (47). In brief, raw sequencing data were demultiplexed using the demux plugin, followed by primer removal with the cutadapt plugin (48). Quality control, including denoising, merging, and chimera removal, was conducted with the DADA2 plugin (49). Non-singleton amplicon sequence variants were aligned using Mafft (50) and utilized to construct a phylogenetic tree with FastTree2 (51). Taxonomic classification of ASVs was performed using the classify-sklearn Naïve Bayes taxonomy classifier within the feature-classifier plugin (52), based on the SILVA database (Release 138) (53).

## Bioinformatics and statistical analysis

All data analyses in this study were primarily conducted using R (version 4.4.0) and Python (version 3.8.8), with the random seed set to 8 for all stochastic procedures to ensure reproducibility. Stacked bar plots were constructed based on the average relative abundance of specific bacterial phyla and genera, sorted in descending order. The top 10 most abundant phyla/genera were displayed individually, while the remaining phyla/genera were grouped under "Other." The data were further stratified by age groups. Alpha-diversity metrics, including Chao1, observed species, Shannon, Simpson, and Faith's Phylogenetic Diversity (PD Whole Tree), were visualized as violin-box plots and statistically compared using the Kruskal–Wallis test followed by Dunn's post hoc test in R. Alpha-diversity analysis was conducted using rarefied data. Beta-diversity analysis was performed using PERMANOVA and PCoA, based on Bray–Curtis and Jaccard distances, implemented in R. PCoA based on Jaccard distance was conducted using binarized data, while Bray–Curtis distance-based PCoA was performed using relative abundance data. To identify differentially abundant taxa across groups, LEfSe analysis was performed with an LDA threshold of >2.0, using R. LEfSe analysis was also based on relative abundance data. Comparative analyses of specific bacterial phyla and genera across the three age groups were conducted using the Kruskal–Wallis test and Dunn's post hoc test, implemented in Python. The Python and R packages used in the data analyses are as follows: Python: numpy (1.22.4), pandas (2.0.3), matplotlib (3.7.5), seaborn (0.13.2), scipy (1.6.2), scikit-posthocs (0.8.1). R: phyloseq (1.48.0), vegan (2.6.6.1), picante (1.8.2), ggsci (3.2.0), ggplot2 (3.5.1), tidyverse (2.0.0), microeco (1.8.0), magrittr (2.0.3), ggtree (3.12.0), and dplyr (1.1.4).

## Data transformation and SNR calculation

During data processing, binarized and relative abundance data were derived directly from the raw count data, whereas log2 and CLR transformations were performed based on relative abundance data. In the binarization process, 1 indicated the presence of an ASV and 0 represented its absence. To address the issue of zero values during log2 and CLR transformation, a small pseudocount of 1e-6 was added. Detailed code and data are available at https://github.com/Z-yuxiang00/Oral_Microbiome-Age/tree/main/2_SNR_codes.

To assess the discriminative power of each microbial feature (e.g., ASVs) across age groups, we computed the SNR based on the principle of one-way analysis of variance (ANOVA). The SNR was calculated as:

$$\text{SNR} = \frac{\sum_{i=1}^{k} n_i(\bar{x}_i - \bar{x})^2/(k-1)}{\sum_{i=1}^{k}\sum_{j=1}^{n_i}(x_{ij} - \bar{x}_i)^2/(n-k)}$$

where $n_i$ is the number of samples in the $i$th age group; $\bar{x}_i$ is the mean of the feature in the $i$th age group; $\bar{x}$ is the overall mean of the feature across all samples; $x_{ij}$ is the value of a microbial feature for the $j$th sample in the $i$th age group; $n$ is the total number of samples; and $k$ is the number of age groups.

The numerator quantifies the between-group variance (signal), while the denominator quantifies the within-group variance (noise). A higher SNR indicates that the microbial feature varies more substantially between age groups than within them and thus may better reflect age-related microbial dynamics.

## Feature selection and model construction

To evaluate model performance, 150 samples were randomly split into a training set ($n = 135$) and an independent validation set ($n = 15$) at a 9:1 ratio. During the selection of the validation set, only samples from ages with two or more individuals were eligible for random sampling (e.g., if there were two or more samples aged 20 years, one could be randomly selected, whereas an age like 19 years with only one sample would not be considered), and for age intervals 20–29, 30–39, 40–49, 50–59, and ≥60 years, three samples were randomly drawn from each. Random sampling was performed in Python (version 3.8.8) with the random seed set to 8 to ensure reproducibility.

Subsequently, LASSO regression was applied to the training set samples ($n = 135$) using four types of transformed data (binarized, relative abundance, CLR, and log2-transformed data) to identify predictive ASVs. The analysis was conducted using the glmnet package (version 4.1.8) in R, with the optimal regularization parameter (λ) determined via 10-fold cross-validation. To enhance the robustness of feature selection, the procedure was repeated 100 times with different random seeds (1–100). ASVs that were selected in ≥50% of the runs were considered stable features for age prediction. These stable ASVs identified from the 135 training samples were then applied to the independent validation set ($n = 15$).

Following feature selection, hyperparameter tuning of models was conducted based on the 135 training samples and the selected predictive ASVs. Considering computational constraints and the ensemble model design, the age bin width was fixed at 20 years. Eight machine learning algorithms were individually tuned as follows: for KNN, a comprehensive grid search optimized the number of neighbors (1, 3, 5, 7, 9), weighting schemes (uniform and distance-based), and distance metrics (Euclidean, Manhattan, Minkowski with $P = 1$ or 2), resulting in 40 hyperparameter combinations. For LDA, two key hyperparameters were tuned: solver method (svd, lsqr, eigen) and shrinkage (none or auto, where applicable). Invalid combinations were excluded, yielding five valid configurations. LR tuning involved three regularization penalties (L1, L2, Elastic Net) across multiple solvers (liblinear, saga, lbfgs, sag, newton-cg), five l1_ratio values

for Elastic Net (0.1, 0.3, 0.5, 0.7, 0.9), and four regularization strengths (C = 0.01, 0.1, 1, 10). Maximum iterations were set to 10,000 for convergence. Gaussian Naïve Bayes tuning focused on the var_smoothing hyperparameter with nine logarithmically spaced values from $1 \times 10^{-9}$ to $1 \times 10^{-1}$ to prevent numerical issues. For the neural network, a two-stage hyperparameter tuning was performed. Stage one involved single hidden layer networks with neurons varying from 10 to 100 (step size = 10), combined with two activation functions (relu, tanh), three learning rates (0.01, 0.005, 0.001), and three L2 regularization strengths ($\alpha$ = 0.0001, 0.001, 0.01), resulting in 18 configurations per neuron count. Fine-tuning was done around the best neuron count using a finer resolution of $\pm$ 10 with step size = 1. Stage two introduced a second hidden layer. Based on the best-performing neuron count from the coarse search in stage one (with step size 10), five first-layer sizes were considered: the initial best and the initial best $\pm$ 10 and $\pm$5. For each of these candidates, the second-layer size was set to 30%–80% of the first layer, using integer values in steps of 5. RF tuning involved a grid search over 432 hyperparameter combinations, including number of trees (50, 100, 200, 300, 400, 500), max depth (5, 10, 20, None), min samples split (2, 5, 10), min samples leaf (1, 2, 4), and max features (sqrt, log2). Bootstrap sampling was enabled, and the random state was fixed at 8. SVM tuning used three kernels (linear, rbf, polynomial), regularization hyperparameter C (0.01, 0.1, 1, 10), and kernel coefficient $\gamma$ (scale or auto), resulting in 24 combinations. For XGBoost, a grid search was performed over 128 hyperparameter combinations. The tuning covered two values for the number of trees (100, 200), two maximum depths (3, 5), two learning rates (0.05, 0.1), and two values each for $\gamma$ (0, 1), min child weight (1, 3), reg alpha (0, 0.1), and reg lambda (1, 5). Subsample and colsample bytree were fixed at 1.0. Model performance and optimal hyperparameters were evaluated by tenfold cross-validation on the training set, with random seed fixed at 8.

In addition to hyperparameter tuning of the aforementioned models, we further optimized the age bin width specific to the ensemble framework. After determining the optimal hyperparameters for each of the eight machine learning algorithms (with the bin width fixed at 20), we conducted a second stage of optimization to identify the most suitable age bin width for each algorithm. Specifically, we evaluated age bin widths ranging from 1 to 64 using tenfold cross-validation on the same 135 training samples. For each algorithm, 64 MAE values were obtained on the training set, and the bin width corresponding to the lowest MAE was selected as the optimal bin width. This process yielded eight fully optimized models, each comprising the algorithm's best hyperparameters along with its corresponding optimal bin width. These eight models were subsequently evaluated on the independent validation set, and the one achieving the lowest MAE was selected as the final model for downstream application.

Among the eight machine learning algorithms, five (XGBoost, neural network, KNN, SVM, and RF) were additionally employed to construct traditional regression models for performance comparison with the ensemble models. For each of the four data types, the procedures of feature selection, hyperparameter tuning, optimal age bin identification, and comparison with traditional models were repeated. All hyperparameter tuning and model construction were performed in Python, with key packages including numpy (1.22.4), pandas (2.0.3), scikit-learn (0.24.1), and xgboost (2.0.3).

## ACKNOWLEDGMENTS

This work was funded by the Natural Science Foundation of Sichuan Province, grant number 2024NSFSC0529, and sponsored by the National Natural Science Foundation of China, grant number 82371897.

Y.Z., F.S., and B.Y.X. contributed to the design. Y.Z. and B.X. contributed to the draft of the manuscript, software, and formal analysis. Y.Z., Y.W., and S.W. contributed to the conceptualization. Y.Z., S.W., X.W., M.L., and F.S. contributed to the methodology. Y.Z., B.X., X.W., B.L., Y.Y., C.W., and C.Z. contributed to the data curation. Y.X.Z. and B.L. contributed to the visualization. Y.Z., Y.W., B.X., and Z.Z. contributed to the visualization investigation.

H.L., F.S., and Y.W. contributed to the resources and supervision. F.S., H.L., and S.W. contributed to the revision.

## AUTHOR AFFILIATIONS

[1]Department of Forensic Genetics, West China School of Basic Medical Sciences & Forensic Medicine, Sichuan University, Chengdu, China
[2]Laboratory of Molecular Translational Medicine, West China Second University Hospital, Sichuan University, Chengdu, China

## AUTHOR ORCIDs

Yuxiang Zhou  http://orcid.org/0000-0003-3438-2986
Benyang Xiao  http://orcid.org/0009-0004-0902-1622
Feng Song  http://orcid.org/0000-0001-6937-3161
Haibo Luo  http://orcid.org/0000-0002-1028-0507

## FUNDING

| Funder | Grant(s) | Author(s) |
|---|---|---|
| Natural Science Foundation of Sichuan Province | 2024NSFSC0529 | Haibo Luo |
| National Natural Science Foundation of China | 82371897 | Feng Song |

## AUTHOR CONTRIBUTIONS

Yuxiang Zhou, Conceptualization, Data curation, Formal analysis, Investigation, Methodology, Software, Visualization, Writing – original draft | Yanyun Wang, Conceptualization, Investigation, Methodology, Resources, Supervision | Benyang Xiao, Data curation, Formal analysis, Investigation, Software, Writing – original draft | Shuangshuang Wang, Conceptualization, Methodology, Writing – review and editing | Zhirui Zhang, Investigation, Resources | Xindi Wang, Data curation, Methodology | Bo Liu, Data curation, Visualization | Yufei Yang, Data curation | Chuanxu Wang, Data curation | Chengye Zhou, Data curation | Miao Liao, Methodology, Resources | Feng Song, Funding acquisition, Methodology, Supervision, Writing – review and editing | Haibo Luo, Funding acquisition, Resources, Supervision, Writing – review and editing

## DATA AVAILABILITY

All relevant data are within the manuscript and its Supporting Information files. The code used for data processing and age prediction was stored in Github (https://github.com/Z-yuxiang00/Oral_Microbiome-Age.git). The raw sequencing data have been deposited in the NCBI Sequence Read Archive (SRA) under accession number PRJNA1336529.

## ADDITIONAL FILES

The following material is available online.

### Supplemental Material

**Supplemental Material (mSystems01182-25-s0001.docx).** Fig. S1 and S2 and supplemental table captions.
**Supplemental Tables (mSystems01182-25-s0002.xlsx).** Tables S1 to S10.

### Open Peer Review

**PEER REVIEW HISTORY (review-history.pdf).** An accounting of the reviewer comments and feedback.

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
