## [Reviewer comments · mSystems]

Predicting age from binarized human oral microbial data combined with an ensemble of classifiers

Yuxiang Zhou, Yanyun Wang, Benyang Xiao, Shuangshuang Wang, Zhirui Zhang, Xindi Wang, Bo Liu, Yufei Yang, Chuanxu Wang, Chengye Zhou, Miao Liao, Feng Song, and Haibo Luo

Corresponding Author(s): Feng Song, Sichuan University

Review Timeline:

Submission Date:	August 8, 2025
Editorial Decision:	September 29, 2025
Revision Received:	October 7, 2025
Accepted:	October 9, 2025

Editor: Nicole De Nisco

Reviewer(s): Disclosure of reviewer identity is with reference to reviewer comments included in decision letter(s). The following individuals involved in review of your submission have agreed to reveal their identity: Shi Huang (Reviewer #1); aman ashar (Reviewer #3); Michael L. Neugent (Reviewer #5)

Transaction Report:

DOI: <https://doi.org/10.1128/msystems.01182-25>

Re: mSystems01182-25 (**Predicting age from binarized human oral microbial data combined with an ensemble of classifiers**)

Dear Dr. Feng Song:

Thank you for your patience while we completed the peer review process for your manuscript. The reviewer's major concerns have all been addressed. Only minor modifications are needed to address reviewer 1's concerns about figure axis labels and legibility.

Please return the manuscript within 30 days; if you cannot complete the modification within this time period, please contact me.

Revision Guidelines

Sincerely,
Nicole De Nisco
Editor
mSystems

Reviewer #1 (Comments for the Author):

Overall, the authors have thoughtfully addressed the key concerns raised in the initial review, with substantial improvements to the methodology, validation, and interpretability of the study. The incorporation of signal-to-noise ratio (SNR) analyses to justify binarization, the systematic hyperparameter tuning for the ensemble model, and the rigorous comparison with previous work (including statistical significance testing) significantly strengthen the robustness of the findings. The expanded discussion on

biological mechanisms linking microbial shifts to aging, as well as the thorough evaluation of confounding factors like sex, further enhances the manuscript's depth.

One minor but important suggestion remains regarding the figures: For scatter plots and other visualizations, the x-axis and y-axis labels are currently small and could be enlarged to improve readability (i.e., font size). Clear, legible axis labels are critical for readers to quickly interpret the data presented, and this adjustment would enhance the overall clarity of the figures.

With this minor revision to the figures, the manuscript would be well-suited for publication.

Reviewer #2 (Comments for the Author):

The authors addressed all the comments.

Reviewer #3 (Comments for the Author):

-

Reviewer #5 (Comments for the Author):

The work presented here presents a novel approach to predicting human age based on oral microbiome profiles by combining binarized ASV data with an ensemble of XGBoost classifiers. Using saliva samples from 150 individuals aged 6-78 years and a larger external dataset of 2,550 samples, the authors demonstrate that binarized microbial features outperform traditional relative abundance and transformed data in signal-to-noise ratio, feature selection stability, and age prediction accuracy. The ensemble model, which aggregates predictions across 32 classifiers using overlapping age bins, achieved a mean absolute error (MAE) as low as 4.33 years in the 20-59 age range and outperformed a previously published model on external validation. The findings suggest that binarized microbial data, paired with ensemble machine learning, offers a robust, interpretable, and scalable framework for microbiome-based age prediction.

The statistical methodology employed in this study is both rigorous and appropriate for the research aims, combining established microbiome analysis tools with innovative machine learning techniques. The authors use robust non-parametric statistical methods (e.g., Kruskal-Wallis, Dunn's post hoc test, PERMANOVA) for exploratory analysis and implement a thoughtful signal-to-noise ratio (SNR) framework to compare the discriminative power of various data transformations. Feature selection is performed using bootstrapped LASSO regression, and the modeling approach is clearly described, centered on a novel ensemble framework of XGBoost classifiers that vote across overlapping age bins. Model performance is comprehensively evaluated using mean absolute error (MAE), both on an internal validation set and on a large external dataset, with appropriate use of cross-validation and non-parametric significance testing (Wilcoxon signed-rank test) to compare predictive accuracy. Importantly, the authors provide detailed descriptions of hyperparameter tuning, data transformation logic, and model selection criteria.

One of the key strengths of the study is its transparency and reproducibility: the authors have made all code and supporting data publicly available via GitHub, with clear documentation and fixed random seeds to enable replication.

Overall, the statistical framework is sound, clearly communicated, and implemented with reproducibility in mind, making it a strength of the study.

Dear Editor,

Many thanks for your comments on our manuscript entitled “**Predicting age from binarized human oral microbial data combined with an ensemble of classifiers**” (No.: mSystems01182-25). We appreciate the valuable comments from the reviewers for improving our work, and we thank you for giving us a chance to revise our manuscript. According to the editor and reviewers’ comments, we revised our manuscript, and the document has been modified using Microsoft Word Track Changes. Our specific responses (point by point) to the editor and reviewers’ comments are provided below.

Editor:

Response:

We appreciate the editor’s reminder regarding data availability. The raw sequencing data have been deposited in the NCBI Sequence Read Archive (SRA) under accession number PRJNA1336529 and are publicly accessible at <https://www.ncbi.nlm.nih.gov/bioproject/PRJNA1336529>.

The corresponding statement has been included in the Data availability section of the manuscript (see lines 832–834 in the Marked-Up Manuscript for details).

Reviewer #1 (Comments for the Author):

Overall, the authors have thoughtfully addressed the key concerns raised in the initial review, with substantial improvements to the methodology, validation, and interpretability of the study. The incorporation of signal-to-noise ratio (SNR) analyses to justify binarization, the systematic hyperparameter tuning for the ensemble model, and the rigorous comparison with previous work (including statistical significance testing) significantly strengthen the robustness of the findings. The expanded discussion on biological mechanisms linking microbial shifts to aging, as well as the thorough evaluation of confounding factors like sex, further enhances the manuscript's depth.

One minor but important suggestion remains regarding the figures: For scatter plots and other visualizations, the x-axis and y-axis labels are currently small and could be enlarged to improve readability (i.e., font size). Clear, legible axis labels are critical for readers to quickly interpret the data presented, and this adjustment would enhance the overall clarity of the figures.

With this minor revision to the figures, the manuscript would be well-suited for publication.

Response:

We sincerely thank you for the positive and encouraging feedback, as well as for recognizing the substantial improvements made in methodology, validation, and interpretation. We truly appreciate your thoughtful suggestion regarding the figure readability.

In response, we have carefully revised the font sizes of the x-axis and y-axis labels, as well as other relevant text elements across all figures, to ensure better visual clarity and consistency. Specifically, the following modifications have been made:

Figure 1c: Font size of the x-axis and y-axis labels increased from 6 to 15; *p*-value font size increased from 12 to 17.

Figure 1b: Font size of the x-axis and y-axis labels increased from 6 to 16.

Figures 1d and 1e (PERMANOVA plots): Font size increased from 12 to 20.

PCoA plot: Axis label font size increased from 10 to 20, and title font size from 16 to 24.

Figures 2c and 2d: Font size increased from 6 to 12.

Figure 3b: x-axis label font size increased from 6 to 10.

Figures 5a and 5b: Font size increased from 12 to 20.

Figure 5c: Title font size increased from 14 to 16; axis label font size from 12 to 20; *p*-value font size from 11 to 16.

We believe these adjustments have significantly improved the readability and overall visual quality of the figures, aligning with your valuable recommendation.

The revised figures are shown below for reference.

Fig. 1

Fig. 2

a**b**
Fig. 3

Fig. 5

Reviewer #2 (Comments for the Author):

The authors addressed all the comments.

Response:

We sincerely thank you for your positive comments and for recognizing our work. We truly appreciate the time and effort you have devoted to reviewing our manuscript, as well as your encouragement for our study. Your thoughtful evaluation is greatly appreciated and has motivated us to further improve the quality of our work.

Reviewer #5 (Comments for the Author):

The work presented here presents a novel approach to predicting human age based on oral microbiome profiles by combining binarized ASV data with an ensemble of XGBoost classifiers. Using saliva samples from 150 individuals aged 6-78 years and a larger external dataset of 2,550 samples, the authors demonstrate that binarized microbial features outperform traditional relative abundance and transformed data in signal-to-noise ratio, feature selection stability, and age prediction accuracy. The ensemble model, which aggregates predictions across 32 classifiers using overlapping age bins, achieved a mean absolute error (MAE) as low as 4.33 years in the 20-59 age range and outperformed a previously published model on external validation. The findings suggest that binarized microbial data, paired with ensemble machine learning, offers a robust, interpretable, and scalable framework for microbiome-based age prediction.

The statistical methodology employed in this study is both rigorous and appropriate for the research aims, combining established microbiome analysis tools with innovative machine learning techniques. The authors use robust non-parametric statistical methods (e.g., Kruskal-Wallis, Dunn's post hoc test, PERMANOVA) for exploratory analysis and implement a thoughtful signal-to-noise ratio (SNR) framework to compare the discriminative power of various data transformations.

Feature selection is performed using bootstrapped LASSO regression, and the modeling approach is clearly described, centered on a novel ensemble framework of XGBoost classifiers that vote across overlapping age bins. Model performance is comprehensively evaluated using mean absolute error (MAE), both on an internal validation set and on a large external dataset, with appropriate use of cross-validation and non-parametric significance testing (Wilcoxon signed-rank test) to compare predictive accuracy. Importantly, the authors provide detailed descriptions of hyperparameter tuning, data transformation logic, and model selection criteria.

One of the key strengths of the study is its transparency and reproducibility: the authors have made all code and supporting data publicly available via GitHub, with clear documentation and fixed random seeds to enable replication.

Overall, the statistical framework is sound, clearly communicated, and implemented with reproducibility in mind, making it a strength of the study.

Response:

We sincerely thank you for your detailed and constructive feedback. We greatly appreciate your recognition of the novelty, rigor, and transparency of our study, as well as your thorough evaluation of our methodology and findings. Your thoughtful and encouraging comments highlight the strengths of our work and motivate us to continue improving our research.

We are glad that you find the statistical framework, feature selection approach, and ensemble modeling strategy appropriate and clearly communicated. We also appreciate your acknowledgment of the reproducibility efforts, including the availability of all code and data on GitHub. Your positive assessment reinforces the significance of our proposed approach to predicting human age based on oral microbiome profiles.

Thank you again for your valuable time and insightful evaluation of our manuscript.

Re: mSystems01182-25R1 (**Predicting age from binarized human oral microbial data combined with an ensemble of classifiers**)

Dear Dr. Feng Song:

Your manuscript has been accepted, and I am forwarding it to the ASM production staff for publication. Your paper will first be checked to make sure all elements meet the technical requirements. ASM staff will contact you if anything needs to be revised before copyediting and production can begin. Otherwise, you will be notified when your proofs are ready to be viewed.

Cover Image Submissions: If you would like to submit a potential Cover Image, please email a file and a short legend to mSystems@asmusa.org. Please note that we can only consider images that (i) the authors created or own and (ii) have not been previously published. By submitting, you agree that the image can be used under the same terms as the published article. Image File requirements: TIF/EPS, 7.5 inches wide by 8.25 inches tall (at least 2,250 pixels wide by 2,475 pixels tall), minimum 300 dpi resolution (600 dpi preferred), RGB, and no figure elements, e.g., arrows or panel labels. The legend should be a short description of the image, 1-2 sentences recommended. Please download and use this interactive template in Adobe to ensure that your proposed cover image meets our size requirements (<https://journals.asm.org/pb-assets/pdf-text-excel-files/ASM-Interactive-Sizing-Cover-Template-1715689791.pdf>).

Sincerely,
Nicole De Nisco
Editor
mSystems